# Does Very Poor Performance Status Systematically Preclude Single Agent Anti-PD-1 Immunotherapy? A Multicenter Study of 35 Consecutive Patients

**DOI:** 10.3390/cancers13051040

**Published:** 2021-03-02

**Authors:** Valérie Gounant, Michael Duruisseaux, Ghassen Soussi, Sylvie Van Hulst, Olivier Bylicki, Jacques Cadranel, Marie Wislez, Jean Trédaniel, Jean-Philippe Spano, Carole Helissey, Christos Chouaid, Olivier Molinier, Xavier Dhalluin, Ludovic Doucet, José Hureaux, Aurélie Cazes, Gérard Zalcman

**Affiliations:** 1Department of Thoracic Oncology, Bichat Claude Bernard Hospital, APHP, CIC Inserm 1425, Université de Paris, 75018 Paris, France; ghassen.soussi@aphp.fr (G.S.); gerard.zalcman@aphp.fr (G.Z.); 2Respiratory Department, Louis Pradel Hospital, Hospices Civils de Lyon, 69002 Lyon, France; michael.duruisseaux@chu-lyon.fr; 3Université Claude Bernard Lyon 1, 69100 Villeurbanne, France; 4Department of Pneumology, University Hospital of Nîmes, 30900 Nîmes, France; sylvie.VANHULST@chu-nimes.fr; 5Respiratory Disease Unit, Hôpital d’Instruction des Armées Sainte-Anne, 83800 Toulon, France; olivier.bylicki@intradef.gouv.fr; 6Department of Pneumology and Thoracic Oncology, Tenon Hospital, APHP, GRC Theranoscan and Curamus Sorbonne Université, 75020 Paris, France; jacques.cadranel@aphp.fr; 7Centre de Recherche des Cordeliers, Université de Paris, Sorbonne Université, INSERM, TeamInflammation, Complement, and Cancer, 75006 Paris, France; marie.wislez@aphp.fr; 8Oncology Thoracic Unit Pulmonology Department, AP-HP, Hôpital Cochin, 75014 Paris, France; 9Groupe Hospitalier Paris Saint-Joseph, Department of Pneumology, Université de Paris, Sorbonne Paris Cité, Unité INSERM UMR-S 1124, 75014 Paris, France; JTredaniel@ghpsj.fr; 10Department of Medical Oncology, Pitié-Salpétrière Hospital, APHP, Sorbonne Université, 75013 Paris, France; jean-philippe.spano@aphp.fr; 11Clinical Research Unit, Hôpital d’Instruction des Armées Bégin, 94160 Saint-Mandé, France; carole.helissey@intradef.gouv.fr; 12Department of Pneumology, Centre Hospitalier Intercommunal de Créteil, University Paris–Est Créteil (UPEC), CEpiA (Clinical Epidemiology and Ageing), EA 7376-IMRB, 94000 Créteil, France; Christos.Chouaid@chicreteil.fr; 13Department of Pneumology, Centre Hospitalier Le Mans, 72037 Le Mans, France; omolinier@ch-lemans.fr; 14Department of Pneumology and Thoracic Oncology, Calmette Hospital, Centre Hospitalier Universitaire de Lille, 59000 Lille, France; Xavier.DHALLUIN@CHRU-LILLE.FR; 15Department of Oncology, Saint Louis Hospital, APHP, 75010 Paris, France; Ludovic.Doucet@ico.unicancer.fr; 16Department of Pneumology, Pόle Hippocrate, University Hospital of Angers, 49100 Angers, France; jose.hureaux@chu-angers.fr; 17Department of Pathology, Bichat Claude Bernard Hospital, APHP, Université de Paris, 75018 Paris, France; aurelie.cazes@aphp.fr

**Keywords:** non-small cell lung cancer, poor performance status, immunotherapy, nivolumab, brain metastases

## Abstract

**Simple Summary:**

Immunotherapies prolong survival of metastatic non-small-cell lung cancer patients. However, their efficacy in patients with very poor general condition is unknown. Best supportive care is the standard of care for these patients because chemotherapy is more toxic and less effective than for patients with good general condition. Most patients die within 1 to 4 months of diagnosis. Consecutive metastatic non-small-cell lung cancer patients with very poor general condition receiving compassionate immunotherapy were accrued by 12 French thoracic oncology departments, over 24 months. Tolerance was acceptable. Overall, 20% of patients were alive at 1 year, and 14% at 2 years. We feel that our study results might suggest that some patients with a very poor general condition (namely those without brain metastases or heavy smokers) could derive long-term benefit from immunotherapy as salvage therapy. We initiated such a prospective phase 2 trial based on these results, which is a cause for hope.

**Abstract:**

Anti-PD-1 antibodies prolong survival of performance status (PS) 0–1 advanced non-small-cell lung cancer (aNSCLC) patients. Their efficacy in PS 3–4 patients is unknown. Conse- cutive PS 3–4 aNSCLC patients receiving compassionate nivolumab were accrued by 12 French thoracic oncology departments, over 24 months. Overall survival (OS) was calculated using the Kaplan-Meier method. Prognostic variables were assessed using Cox proportional hazards models. Overall, 35 PS 3–4 aNSCLC patients (median age 65 years) received a median of 4 nivolumab infusions (interquartile range [IQR], 1–7) as first- (*n* = 6) or second-line (*n* = 29) therapy. At a median of 52-month follow-up (95%CI, 41–63), 32 (91%) patients had died. Median progression-free survival was 2.1 months (95%CI, 1.1–3.2). Median OS was 4.4 months (95%CI, 0.5–8.2). Overall, 20% of patients were alive at 1 year, and 14% at 2 years. Treatment-related adverse events occurred in 8/35 patients (23%), mostly of low-grade. After adjustment, brain metastases (HR = 5.2; 95%CI, 9–14.3, *p* = 0.001) and <20 pack-years (HR = 4.8; 95%CI, 1.7–13.8, *p* = 0.003) predicted worse survival. PS improvement from 3–4 to 0–1 (*n* = 9) led to a median 43-month (95%CI, 0–102) OS. Certain patients with very poor general condition could derive long-term benefit from nivolumab salvage therapy.

## 1. Introduction

Immune checkpoint inhibitors (ICI) are shown to increase survival in patients with advanced non-small cell-lung cancer (aNSCLC) and good Eastern Cooperative Oncology Group (ECOG) performance status (PS) 0–1, as compared to standard chemotherapy in the second-line setting [1,2,3,4], and as front-line therapy for patients with programmed cell death ligand one (PD-L1) expression in ≥50% tumor cells [5]. Nonetheless, the efficacy of these agents when administered to patients with poor and very poor PS remains largely unexplored in both settings. 

However, the prevalence of poor and very poor PS patients at the time of diagnosis is high in lung cancer patients. Lilenbaum et al. [6] reported a 21% prevalence of PS 2 among lung cancer patients at the time of diagnosis. Prevalence of PS 3 and PS 4 patients was 11% and 1%, respectively. In the French survey KBP-2010-CPHG [7], prevalence of PS 2, PS 3, and PS 4 at the time of diagnosis was 18.4%, 9.9%, and 2.8%, respectively.

For patients with metastatic NSCLC PS 3 or 4, there is no other standard treatment option than best supportive care. The recommendation against chemotherapy for patients with very poor PS dates back to the early 1980s, when PS was found to be a predictor of poor survival, reduced response, and worsened toxicity with regards to chemotherapy available at that time. All trials in unselected patients with poor and very poor PS turned out to produce negative results, with most patients dying within 1 to 4 months of diagnosis. 

The only positive trials conducted in this population with poor general condition are trials focused on patients with addictive oncogenic mutation. In the Phase II trial dedicated to patients with epidermal growth factor receptor (EGFR)-activating mutation that received gefitinib [8], the overall response rate was 66%, and the disease control rate 90%. PS improvement was observed in 79% of patients, with 68% of 22 patients improving from PS ≥ 3 at baseline to PS ≤ 1 on treatment. A Phase II study dedicated to patients with anaplastic lymphoma kinase (ALK) rearrangement and poor PS that were treated with alectinib also showed a high response rate [9]. The median PFS was 16 months (95% confidence interval [95% CI], 7.1–30.8), and median survival time was 30 months (95% CI, 11.5–not reached) [10], again contrasting with historical chemotherapy’s poor results in patients with altered general condition. These trials were instrumental in changing practices so that currently, when such an addictive tumor mutation is detected (EGFR, ALK, or ROS1), the patient should receive a tyrosine kinase inhibitor, regardless of PS. These trials have thus demonstrated the feasibility to treat PS 3–4 patients, within a selected population, with a well-tolerated drug, eventually resulting in a high response rate and fast general condition improvement, as well as quality of life improvement and survival advantage. 

To our best knowledge, no prospective trial that included PS 3 or 4 patients to receive ICI was ever reported to date. To date, only four prospective trials involving PS 2 patients are published, including two Phase II trials, such as CheckMate 171 with nivolumab [11] and PePS2 with pembrolizumab [12], and two Phase III/IV trials, such as CheckMate 153 with nivolumab [13] and CheckMate 817 with both nivolumab and ipilimumab [14]. Patients were included in these trials, irrespective of their PD-L1 tumor expression or tumor mutational burden (TMB). The safety profile was demonstrated to be broadly similar to that observed in PS 2 and PS 0–1 patients. In contrast, while the overall survival (OS) was worse in PS 2 patients compared to the others, there were surprisingly few long-term survivors observed. Based on such data, it is still difficult to decrypt whether this poorer outcome is simply the consequence of the patients’ worse prognosis or if altered PS could actually represent a prognostic factor predicting a worse outcome upon immunotherapy, which is likely associated with an unfavorable immunological status.

We recently observed Lazarus-type responses to anti-PD1 in two NSCLC patients with very poor condition, yet very high PD-L1 expression [15]. These patients improved from PS ≥3 prior to immunotherapy initiation to PS 0, following 1-month anti-PD-1 treatment, with major tumor shrinking and long-term survival over a follow-up period currently exceeding 4 years. Interestingly, patients with poor PS could likewise develop efficacious immune responses upon ICI therapy. This contradicts the current concept, pointing towards an exhausted immune response ability in poor PS patients. In these special cases, we observed a major quality of life improvement and extended survival. If their poor general condition was considered, thereby contra-indicating such salvage immunotherapy in line with current concepts and registration indications, this clinically meaningful benefit would not have occurred. At that time, we were wondering whether our observations could be reproduced. We thus set up a retrospective observational multicenter study (named SAVIMMUNE RETRO) designed to collect clinical features, PD-L1 status, response to therapy, and outcomes in a cohort of advanced NSCLC patients with very poor PS that were treated using salvage nivolumab.

## 2. Materials and Methods

### 2.1. Patient Cohort

SAVIMMMUNE RETRO was designed as a retrospective multicenter observation study. We invited 12 French center investigators that were willing to participate in this survey to collect data from all consecutive advanced (Stage IIIC/IV) NSCLC patients, PS 3 or 4, who received at least one course of salvage nivolumab, within a 24-month time frame ranging from April 2015 to April 2017. Clinical records were extracted and collected locally by investigators, with survival data updated on 16 August 2020. PS at diagnosis and upon treatment was assessed from the patients’ charts, based on the ECOG criteria, given that the selected investigators were deemed familiar with selecting patients for academic (French Intergroup, IFCT) or company-sponsored clinical trials, which require such a systematic evaluation.

The de-identified database was stored in the electronic system of the Early Phase Unit INSERM CIC-1425, University Hospital Bichat, *Assistance Publique-Hôpitaux de Paris*, in compliance with the French regulatory rules to preserve the privacy of patient’s safety, such data electronic files were declared to the *“Comité National Informatique et Liberté”* (CNIL). A written informed consent was obtained from all patients that were still alive at the time of data recording, whereas an information sheet stating the right to modify or oppose data collection was provided by the investigators to each deceased patient’s relatives. 

The study protocol was approved by the Institutional Review Board of the French Society of Respiratory Diseases (SPLF) and was registered under the CEPRO number #2020-061-R1.

### 2.2. Nivolumab Treatment

Nivolumab was administered intravenously at a dose of 3 mg per kilogram of body weight every 2 weeks, according to its first French registration in 2015. The drug was discontinued in the event of progressive disease or unacceptable toxicity, as assessed by the investigator.

### 2.3. PD-L1 Expression

PD-L1 expression of tumor cells was determined using immunohistochemistry (IHC) by the local pathology laboratory, based on its own routine IHC platform with 22C3 (Dako), SP263 (Ventana), 28-8 (Dako), or E1L3N (Cell Signaling Technology) antibody, all approved in France for such purpose. The results were not systematically known at the time of nivolumab treatment initiation. Considering the study’s retrospective design, no centralized testing was performed.

### 2.4. Primary Objective

The primary study objective was OS at 1 year, following the first nivolumab prescription, either in first-line or subsequent lines of treatment in PS 3–4 patients, who were re-enrolled in the nivolumab compassionate use program.

### 2.5. Secondary Objectives

Secondary objectives included overall response rate (ORR), defined according to the Response Evaluation Criteria in Solid Tumors (RECIST), Version 1.1, progression-free survival (PFS), and PS improvement rate (assessed according to the Eastern Cooperative Oncology Group criteria). The PS improvement rate was defined as the proportion of per-protocol patients whose baseline PS improved upon nivolumab treatment.

Incidence and severity of TRAEs were recorded and graded according to the National Cancer Institute Common Terminology Criteria for Adverse Events Version 4.0.

### 2.6. Statistical Analysis

OS was calculated from the date of first immunotherapy administration to death from any cause. PFS was calculated from the date of first nivolumab administration to the date of proven disease progression or death from any cause.

Comparisons between patient characteristics were performed using Chi-squared or Fisher’s exact test for discrete variables. The Wilcoxon matched-pairs signed-rank test was applied to compare ordinal variables between paired observations. Survival analyses were performed using the Kaplan–Meier method, while the log-rank test was applied to compare survival distributions between groups. The HRs and their respective 95% CI were calculated from the univariable analysis using Cox regression. The follow-up duration was evaluated using the reverse Kaplan–Meier method. All hypothesis tests were two-tailed, with *p* values inferior to 0.05 indicative of statistical significance.

The statistical plan consisted of analyzing the impact of demographical and clinical-pathological characteristics on disease response and survival outcomes, based on the following five features—age, PS, smoking status, brain metastases, and liver metastases (at least 6 events for each variable) in an effort to avoid model over-fitting. An expanded exploratory univariate analysis was performed post-hoc, including PD-L1 status, histo- logy, K-Ras status, first-line therapy, best PS observed upon ICI treatment, response according to RECISTv1.1 criteria, and treatment-related adverse events (TRAEs). 

The multivariable analysis was conducted using backward stepwise Cox regression modeling, with OS as the dependent variable and prognostic factors as the explanatory variables. Variables whose value was known prior to the nivolumab treatment (thus excluding response to nivolumab, TRAEs, and PS change from baseline) were included in the final model, according to their statistical significance in the univariate analysis (cutoff, *p* = 0.10).

Statistical analyses were performed using IBM SPSS Statistics for Windows, Version 25.0 (IBM Corp., Armonk, NY, USA) and GraphPad Prism Version 8.4.3 for Windows (GraphPad Software, San Diego, CA, USA). Graphics were rendered using the latter tool, in addition to Microsoft Excel Version 2013 for Windows (Microsoft Corporation, 2013), and RStudio Version 1.3.1093 (Integrated Development for R. 2009–2020 RStudio, PBC, Boston, USA).

## 3. Results

### 3.1. Patient Characteristics

During the 24-month study period, 35 consecutive NSCLC patients with very poor PS and treated with nivolumab as salvage therapy were included by 12 academic centers. Patients’ baseline characteristics are shown in Table 1. Of these 35 patients (23 men), 29 (83%) exhibited PS 3 and six (17%) showed PS 4. Median patient age was 65 years (IQR, 62–74). The majority of patients (32/35) were active smokers, yet 3/35 were never-smokers. Most patients (23/35) displayed adenocarcinomas with Stage IV disease (34/35), whereas the remaining patients exhibited IIIC Stage with bulky disease. Overall, 35% patients had at least three metastatic sites, while 29% presented brain metastases and 26% liver metastases. Most patients received nivolumab following platinum-based chemotherapy, 20/35 of whom were in second-line, and 9/35 were in third-line or more. Due to their poor general condition, there were 6/35 patients that received frontline salvage nivolumab, which was not yet registered for such indication, despite being the only immuno-oncology drug then available in France. Response evaluation following platinum-based chemotherapy as first-line was available for 29 patients, with evaluable response in 25, showing disease progression in 8/25 (32%), stable disease in 6/25 (24%), and partial response in 11/25 (44%). These latter patients progressed at a later time-point.

Tumor cell PD-L1 IHC expression, as determined by the local pathology laboratory, was not systematically performed when nivolumab treatment was initiated. Therefore, patients were not selected according to their PD-L1 tumor expression in most centers. In only one case, the remaining tumor material was insufficient, and the analysis was thus not possible. Of the 34 patients whose tumor samples were assessable for PD-L1 expression, 11 (32%) showed no tumor PD-L1 expression, whereas 23 (68%) exhibited PD-L1 tumor expression in at least 1% of tumor cells, including 14 patients (41%) with high PD-L1 tumor expression in at least 50% of tumor cells. 

### 3.2. Outcomes in the Global Population

Median follow-up for OS was 52 months (95% CI, 41–63). At time of analysis, 32 (91%) patients had died, while three (8%) were still alive. No patients were lost to follow-up. Median OS was 4.4 months (95% CI, 0.5–8.2), with 1-year OS at 20% and 2-year OS at 14% (Figure 1). Early mortality in the 30- and 60-day periods following nivolumab initiation occurred in 17% and 29% of patients, respectively. Disease progression was the most common reason for death (81%).

A median four (IQR, 1–7) of nivolumab doses was administered. At data cut-off, two patients were still on treatment. Disease progression was the most common reason for drug discontinuation (54%), followed by death (21%), and then toxicity (12%). Nine patients (26%) were not evaluable for RECISTv.1.1, including one patient with no measurable target lesion and eight who died before disease assessment. For 26 patients evaluable for RECIST v.1.1, 11/26 (42.3%) achieved a partial response, 5/26 (19.2%) had a stable disease as best response, with 10/26 (38.5%) displaying progressive disease. Partial response rate in the intent-to-treat analysis was 32% for the whole series (Appendix A), whereas in patients with PD-L1 tumor expression ≥1%, partial response rate was 39%. No statistically significant differences in objective response were found among PD-L1 subgroups using cutoff values of 1% (Appendix A) or 50% (no showed). Immunotherapy response median duration in evaluable patients was 16 months (IQR, 46–3) (95% CI, 0–40); individual response durations were plotted in Figure 2, with 7/35 (20%) patients showing response durations that exceeded 9 months. Median PFS was 2.1 months (IQR, 5.7–1.1) (95% CI, 1.1–3.2).

TRAEs as assessed by investigations that occurred in 8/35 patients (23%). Most events were of low grade and consisted of thyroid or skin disorders. Two patients experienced treatment-related pneumonitis, Grade 2. Grade 3–4 TRAEs occurred in one patient who experienced disseminated intravascular coagulation and encephalitis, following the first nivolumab perfusion. These adverse events were resolved following high-dose corticosteroids, with a partial response observed during 5 months in this patient. TRAE Grade 5 was observed after 2 months of nivolumab therapy in another patient, who presented acute respiratory distress with a “white-out lung” appearance. No histological formal diagnosis could be established.

In univariate analysis that considered only five pre-specified variables to ensure correctly powered analysis, three factors were significantly associated with a lower OS—non-smokers and smokers <20 PY, brain metastases, and liver metastases, with results presented in Table 2 (the expanded analyses are provided in Appendix A). Median OS in the group of never-smokers and smokers <20 PY was 1.6 months (95% CI, 0–4.6) versus 5.7 months (95% CI, 0.8–10.5) in the heavy-smoker group (≥20 PY) (*p* = 0.01). Median OS in patients with brain metastases was 2.1 months (95% CI, 0.6–3.6) versus 8 months (95% CI, 0.4–15.6) in those without brain metastases (*p* = 0.003) (Figure 3a). Median OS in patients with liver metastases was 2.1 months (95% CI, 0.5–3.8) versus 7.3 months (95% CI, 1.7–12.9) in those without liver metastases (*p* = 0.047) (Figure 3b). In univariate analysis, no statistically significant differences in OS were found among PD-L1 subgroups (*p* = 0.67). Notably, patients with tumor PD-L1 expression higher than 50% did not specifically derive any survival advantage (median OS = 4.6 months) as compared to patients showing 1–49% PD-L1 positive tumor cells (median OS = 7.2 months). Nevertheless, patients with PD-L1 tumor expression less than 1% displayed a shorter median OS (2.3 months) (Appendix A).

After adjusting for variables that demonstrated a *p*-value ≤0.1 in univariate analysis, only brain metastases (HR = 5.2; 95%CI, 1.9–14.3, *p* = 0.001) and smoking status (<20 PY) (HR = 4.8; 95%CI [1.7–13.8], *p* = 0.003) were still independent prognostic factors that were significantly associated with worse OS (Table 2). 

As expected, partial response to nivolumab was associated with a longer 23-month (95% CI, 0–58) OS, as compared to 8 months (95% CI, 6–10) in patients with stable disease, and 2.3 months (95% CI, 0.6–4]) in those with disease progression (*p* < 0.0001) (Appendix A).

Interestingly, TRAEs occurring upon nivolumab therapy were significantly associated with longer OS (HR, 0.38; 95% CI, 0.2–0.9, *p* = 0.03). OS was 10.7 months (95% CI, 10.3–11.2) in patients with TRAEs versus 2.3 months (95% CI, 0.6–4) in those without any observed TRAEs (Appendix A). 

During nivolumab therapy, 14/35 (40%) patients experienced ECOG PS improvement (*n* = 9 PS 0–1 (26%); *n* = 5 PS 2 (14%)) (Figure 4). Median OS in patients with PS improvement to PS 0 upon nivolumab therapy was NR versus 23.2 months (95% CI, 0.6–2) in those whose best PS was 1 versus 7.3 months (95% CI, 7.1–7.5) in those whose best PS was 2, but only 2.1 months (95% CI, 1.3–2.9) in those that remained in PS 3 (*p* < 10^−6^) (Appendix A).

## 4. Discussion

In our series of unselected NSCLC PS 3–4 patients that received nivolumab as salvage therapy, OS was 4.4 months. In a published retrospective, non-comparative series involving PS 3–4 patients that only received best supportive care, the reported OS was 2.4 months [16]. In the absence of any face-to-face prospective comparison, it is hard to decrypt—based on such median OS data—whether immune-oncological drugs could actually overcome the natural history of cancer disease in such patients. In our study, OS was actually only 2 months in patients with disease progression upon nivolumab versus 23 months in those who experienced a partial response under nivolumab.

In our series, nivolumab was mostly administrated in second- or subsequent-line settings, notably in 29/35 patients. In the second-line setting registration trials, CheckMate 017 [2] and CheckMate 057 [1], such trials only accrued PS 0–1 patients. In our study, 26% of patients received nivolumab beyond second-line, whereas 29% presented brain metastases, versus only 6% in CheckMate 017 trial [2], with all previously treated in the latter. Interestingly, Kanai et al. [17] reported exacerbation of neurologic symptoms in 58% of patients with central nervous system metastases, which resulted in treatment discontinuation, mainly in PS 3 patients. 

Conversely, in our series, six patients (17%) received nivolumab in the first-line setting, irrespective of PD-L1 tumor expression, and thus outside of the guidelines and Food and Drug Administration (FDA)/European Medicines Agency (EMA)-approved indications, and prior to the CheckMate 026 negative trial being published [18]. This latter trial randomized frontline chemotherapy versus frontline nivolumab in aNSCLC PS 0–1 patients, irrespective of their PD-L1 tumor expression. Such treatment decision was mainly supported by poor PS precluding any chemotherapy, while often being encouraged (though not always) by high PD-L1 tumor expression, in the absence of any available alternative treatment, except for best supportive care.

PS is the major prognostic factor predicting OS, irrespective of the administered treatment [19]. Patients with poor and very poor PS actually exhibit a significantly worse survival compared to those with PS 0–1. Small Phase II trials included PS >2 patients, with the aim of evaluating chemotherapy in such a poor PS population. For instance, the IFCT 0301 Phase II trial compared three treatment strategies in chemotherapy-naive unselected aNSCLC patients—PS 2–3 patients [20]. In the PS 2 patient subgroup, median OS times were 3.0 months (95% CI, 1.9–4.1) in the gefitinib arm, 3.1 months (95% CI, 2.0–6.4) in the gemcitabine arm, and 6.6 months (95% CI, 3.5–8.3) in the docetaxel arm. In contrast, in the PS 3 patient subgroup, median OS times were 1.9 months (95% CI, 0.8–2.2), 1.8 months (95% CI, 0.9–4.3), and 1.1 months (95% CI, 0.6–2.7), respectively. The same observation was made in other trials that included PS 3 patients [21,22,23]. PS 2 patients were significantly more likely to respond to therapy than PS 3 [22]. Early mortality in the 60-day period following chemotherapy initiation was high, amounting to 38.5% in the Phase II trial with gemcitabine given as single drug therapy in the first-line setting for patients with Karnofsky ≤70% [24]. In our study, early mortality within the first 60 days was 29%. As our patients were PS 3–4 and because most of them received nivolumab in second- and subsequent-line settings, such results are not unexpected; indeed, such data do not compare unfavorably with chemotherapy-induced results. Actually, disease progression was deemed to be the major cause of death, rather than toxicity-related adverse events. In the current study, PS 3 patients (*n* = 35) exhibited a median 4.4-month OS, as compared with 1.1 months for PS 4 patients (*n* = 6), yet this difference did not reach statistical significance, which is likely due to the limited sample size. These results once more confirm a critical role for PS in survival. 

In our study, 23% of patients experienced TRAEs, with one TRAE Grade 3–4 and one TRAE Grade 5. To date, only few data on ICI safety are available in such very poor PS patients. Thus far, we only have scarce information available about ICI tolerance in PS 2 patients. In CheckMate 171 [11], the incidence of TRAE Grade 3–4 was 7% for the PS 2 population, and 12% for the overall population (including PS 0–1 patients and ≥70-year-olds). In CheckMate 153 [13], the incidence of TRAE Grade 3–4 was 9% for the PS 2 population, and 6% for the overall population. In a retrospective single center study [25], toxicity was the reason for ICI discontinuation in 24% of PS 0–1 patients, but in only 6% of PS ≥2. No significant PS-related differences were found among patients that were hospitalized for toxicity, those that received steroids for immune-related toxicity, and those that experienced toxicity-related death. In PS 0–1 patients, single agent anti-PD1 treatment were reported to also induce substantial rates of grade 3–4 TRAEs in the first-line setting, from 29.9% and 18.8% in the Keynote-024 [5] and Keynote-042 [26] phase 3 trials, respectively, leading to 1.3 and 2% of deaths, respectively, to 33% in the CheckMate-227 [27] phase 3 trial with both anti-PD1 and anti-CTLA4 antibodies, leading to 1.4% of deaths.

As previously suggested in retrospective studies [28,29], TRAE occurrence was associated with longer OS despite some TRAEs that were found to be associated with nivolumab discontinuation. Taken together, these observations would not necessarily suggest a poorer safety profile in lower PS patients, with most immune-related AEs being reversible, provided that active treatment like corticosteroid therapy is given early. However, the main limitation of our retrospective study is the lack of sequential and longitudinal evaluation of the patients’ quality of life data, using validated assessment scales.

In contrast, PD-1 inhibitors were clearly shown to be effective in several poor PS patients. Nevertheless, it appears evident that only a small PS 3–4 patient subset could draw definite benefit from such treatments. In our study, only one-fifth of patients (20%) were alive at 1 year, whereas three were still alive at ≥4 years. Although no dedicated PS 3 patient trials are currently ongoing with ICI, we previously reported several Lazarus-type anti-PD-1 responses in NSCLC patients in a very poor condition, yet with very high PD-L1 expression [15]. Likewise, several authors already suggested that pembrolizumab could be considered in critically-ill NSCLC patients exhibiting a PD-L1 expression ≥50% [30]. However, we failed to observe such a favorable role of high PD-L1 expression, possibly on account of the PD-L1 IHC assay’s versatility, using different antibody clones and IHC platforms [31], or because of the limited sample size. The main challenge remains to be able to identify, within the poor or very poor PS population, those elective patients that could actually derive ICI benefits. For instance, in the PePS2 trial [12] that was focused on pembrolizumab-treated PS 2 patients, median OS was 14.6 months (4.6–NR) for patients with PD-L1 tumor proportion score ≥50%; thus, PD-L1 tumor expression remains a major prognostic or predictive factor. As previously reported, the PD-L1 biomarker was not 100% exact, as this biomarker failed to predict long-term survival in two of our patients, whereas one patient with PD-L1 negative tumor, conversely, did respond to nivolumab over a 9-month treatment. Taken together, these observations demonstrated this biomarker’s versatility, which could also explain, in addition to our limited sample size, why PD-L1 status was not found to be associated with OS. Beyond the lack of power in our study and the technical issues with PD-L1 immunostaining, we cannot exclude either that more complex biological factors could interfere with PD-L1 tumor expression, such as resident T-cell infiltration [32] and cancer-associated fibroblast inhibitory effect of T-cell mediated anti-tumor activity [33], while gut microbiome could also play a major role in NSCLC patients [34].

Receiving ICI in the last 30 days of life was reported to be associated with poor end-of-life quality, with fewer hospice referrals and more in-hospital deaths [25,35]. Therefore, ICI prescription should not deprive patients of palliative care and time for “life–death transition process”. Given this context, it must be noticed that in the Petrillo et al. [25] study, PD-L1 expression was unknown in 67% of tumors, whereas there were only 6% tumor patients with high PD-L1 expression. For these very poor PS patients, we strongly believe that the right patient selection for ICI administration proves to be mandatory, with PD-L1 tumor expression analysis being at least required, even though this biomarker assessment is still imperfect. Based on our reported results, immunotherapy should be discouraged in PS 3–4 patients with brain metastases. Moreover, other putative predictive biomarkers would deserve prospective evaluation, notably several patient characteristics that reflect inflammatory and nutritional status, including the lung immune prognostic index, body mass index, or derived neutrophil-to-lymphocyte ratio. These factors could facilitate the physicians’ decision-making process as to whether ICI could have a role to play as salvage therapy in PS 3–4 patients.

Our study has several limitations. First, even if all consecutive patients should have been accrued by the 12 participating centers, its retrospective design could have led to a selection or recall bias. Moreover, its sample size was limited, possibly suggesting that our experienced investigators scrupulously followed the ICI-registered mentions for only including PS 0–1 patients. One should be aware that our results should not be over-interpreted and lead to over treatment in advanced poor PS NSCLC patients, with all the inherent consequences, mainly adverse events and economic burden for patients and families, taking into account the scarce available prospective data in such patients.

Second, the major factors conditioning ECOG PS 3–4, such as precise disease burden or comorbidities, were not assessed in the current study. Third, only PD-L1 expression was employed as a predictive marker for anti-PD-1 therapy response and survival; therefore, we were not able to study TMB, which is not routinely measured in France. TMB could possibly explain the influence of smoking in our multivariate analysis, since heavy smokers are usually thought to exhibit higher TMB.

## 5. Conclusions

Based on these retrospective data that should only be considered as hypothesis generation, and taking into account the study’s above-mentioned limitations, the French Collaborative Thoracic Intergroup (IFCT) launched a prospective, single-arm, first-line Phase 2 study, SAVIMMUNE (IFCT 1802, Eudract N°: 2018-004742-42, NCT04108026). This study’s primary aim is to assess the anti-PD-L1 monoclonal antibody durvalumab in patients with high PD-L1 tumor expression (≥25% tumor cells) and poor general condition. Two sequential strata are scheduled, with only one including PS 2 patients. If no safety signals are noted, a second strata devoted to PS 3 patients will be investigated, with safety parameters as the primary study endpoint.

## Figures and Tables

**Figure 1 cancers-13-01040-f001:**
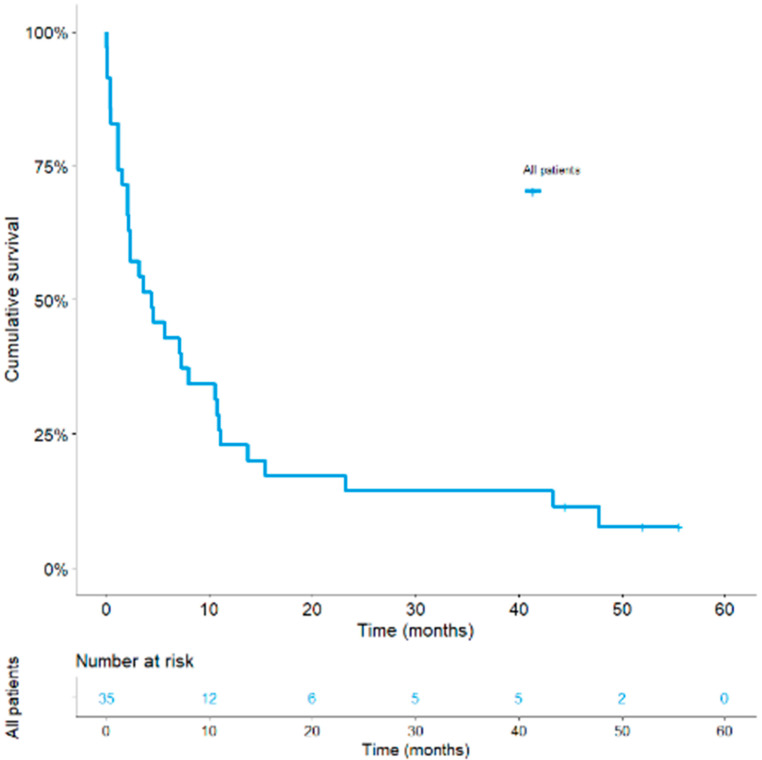
Kaplan–Meier curves of overall survival in the global population.

**Figure 2 cancers-13-01040-f002:**
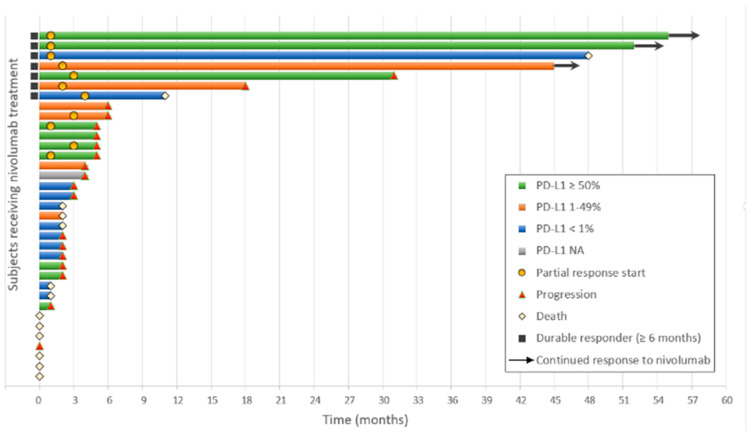
Swimmer plot of tumor response for subjects receiving nivolumab, by month. PD-L1, programmed death-ligand 1.

**Figure 3 cancers-13-01040-f003:**
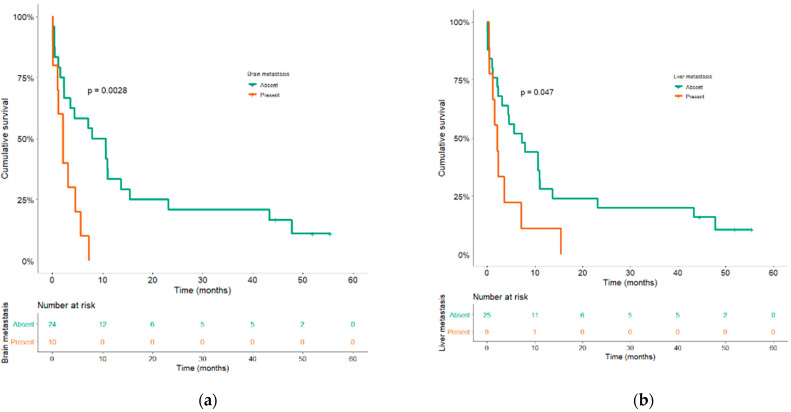
Kaplan–Meier curves of overall survival according to brain (**a**) and liver metastases (**b**).

**Figure 4 cancers-13-01040-f004:**
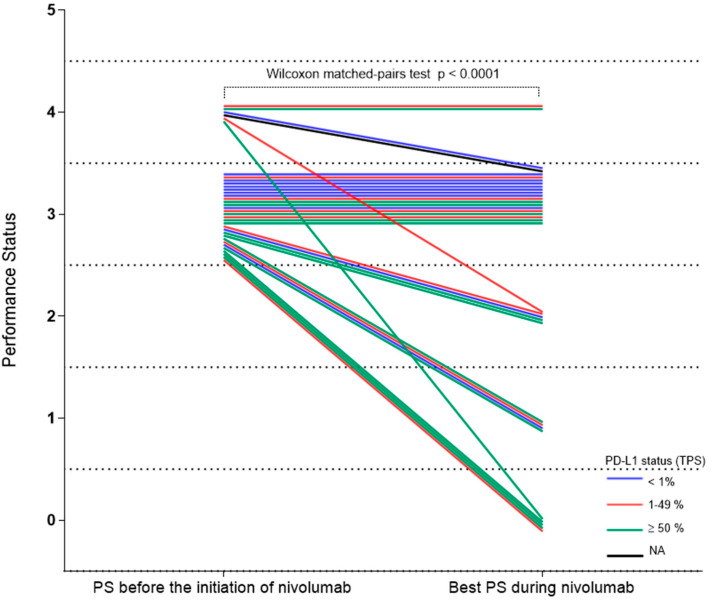
Change of performance status upon nivolumab. Change of performance status for each patient during nivolumab treatment, by PD-L1 status. PD-L1, programmed death-ligand 1; PS, performance status.

**Table 1 cancers-13-01040-t001:** Baseline Patient Characteristics.

Characteristics (N = 35)	N (%)
**Age, years**	
Mean	67
Median (IQR)	65 (62–74)
**Gender**	
Male	23 (66)
Female	12 (34)
**PS ECOG**	
PS 3	29 (83)
PS 4	6 (17)
**Smoking status**	
Current and Former smoker	32 (91)
Never-smoker	3 (9)
**Histology**	
Adenocarcinoma	23 (66)
Squamous cell carcinoma	7 (20)
Other	5 (14)
**PD-L1 status (TPS)**	
0%	11 (32)
1–49%	9 (27)
≥50%	14 (41)
Unknown	1
**Molecular status**	
K-Ras mutation	11 (35)
BRAF V600E mutation	1 (3)
MET exon-14 mutation	1 (3)
Other	2 (6)
EGFR mutation	0
ALK rearrangement	0
No mutation or rearrangement	17 (53)
Unknown	3
**Number of metastatic sites**	
<3	22 (65)
≥3	12 (35)
Unknown	1
**Brain metastasis**	
Present	10 (29)
Absent	24 (71)
Unknown	1
**Liver metastasis**	
Present	9 (26.5)
Absent	25 (73.5)
Unknown	1
**Treatment line**	
1st line	6 (17)
2nd line	20 (57)
3rd line or later	9 (26)

**Table 2 cancers-13-01040-t002:** Univariate (Kaplan–Meier) and Multivariate Analysis (Cox proportional hazards) results.

		Univariate Analysis					Multivariate Analysis			
Variables	N	OS (mo)	95% CI	HR	95% CI	*p* Value	Wald	aHR	95% CI	*p* Value
Age (years) < 65 ≥ 65	35 16 19	5.7 2.1	0.4–11.0 1.4–2.9	- 1.8	0.9–3.7	0.092				
Smoking status (PY) ≥ 20 < 20	34 27 7	5.7 1.6	0.8–10.5 0–4.6	- 2.9	1.2–7.0	0.014	8.6	- 4.8	- 1.7–13.8	0.003
PS before nivolumab initiation 3 4	35 29 6	4.4 1.1	1.9–6.9 0–9.3	- 1.0	0.4–2.7	0.941				
Liver metastasis No Yes	34 25 9	7.3 2.1	1.7–12.9 0.5–3.8	- 2.2	1.0–5.1	0.047				
Brain metastasis No Yes	34 24 10	8.0 2.1	0.4–15.6 0.6–3.6	- 3.5	1.5–8.5	0.003	10.4	- 5.2	- 1.9–14.3	0.001

OS, overall survival; 95% CI, 95% confidence interval; HR, hazard ratio; aHR, adjusted hazard ratio; PS, Eastern Cooperative Oncology Group performance status; and PY, pack-years.

## Data Availability

The data presented in this study are available on request from the corresponding author.

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
