# Peer review of "Does Very Poor Performance Status Systematically Preclude Single Agent Anti-PD-1 Immunotherapy? A Multicenter Study of 35 Consecutive Patients"

_cancers, 2021, doi:10.3390/cancers13051040_

Round 1

Reviewer 1 Report

The manuscript entitled "Does very poor performance status systematically preclude single agent anti-PD-1 immunotherapy? A multicenter study of 35 consecutive patients" presented the therapeutic efficacy (as a salvage therapy) of nivolumab in the NSCLC patients who had poor performance status (PS 3-4). This is a retrospective non-comparative study. Due to the research design and limited patient number, a solid conclusion cannot be made. However, for the NSCLC patients who are at late stage, have poor performance status, are not suggested to receive further constructive therapy, but still struggle to survive, this study provides important information for the clinicians and the patients. However, we should be very careful to discuss this issue, because this kind of study may bring some debate in over-treatment and economic burden for the patients and their families. In fact, the Discussion section may be the most important part of this manuscript, and the authors provided sufficient references of related  researches and gave a fair comment in the Discussion. This reviewer suggests that the concern about over-treatment and economic issue may be included in the last part of the Discussion.

Reviewer 2 Report

The study aimed to assess the anti PD-L1 monoclonal antibody in patients with high PD-L1 expression and with poor general condition (performance status [PS] 3-4). Though the sample size of the study is small (n=35), the data provided a scope for usage of nivolumab as a salvage therapy for advanced NSCLC patients with poor performance status. It wasconducted in advanced NSCLC patients with PS 3-4 which is first of its kind. Nivolumab improved the OS from earlier 2.4 months to 4.4 months in PS3 patients. Overall, this study can help physicians’ decision-making regarding whether ICI could be used as a salvage therapy in PS 3-4 patients.

This reviewer has the following concerns.

  1. The study appears to show that nivolumab only benefited patients of poor general condition with no brain metastasis or the patients who are not heavy smokers (less than 20 pack-years). The reason underlying this result should be properly discussed. Is this related to overall mutational burden and comorbidities of the patients ?
  2. The relation between PD-L1 expression levels in the tumors and PS-3 or PS-4 status of patients were not established. Why PD-L1 status was not associated with the overall survival (OS) rate is not discussed.
  3. More information should be provided for treatment related adverse events (TRAEs). Are TRAEs in PS3-4 patients different from that reported in PS0-1 patients?
